# Genomic regions of current low hybridisation mark long-term barriers to gene flow in scarce swallowtail butterflies

**Sam Ebdon**[1]*, **Dominik R. Laetsch**[1], **Roger Vila**[2], **Stuart J. E. Baird**[3☯], **Konrad Lohse**[1☯]

**1** Institute of Ecology and Evolution, The University of Edinburgh, Edinburgh, United Kingdom, **2** Institut de Biologia Evolutiva (CSIC-Universitat Pompeu Fabra), Barcelona, Spain, **3** Institute of Vertebrate Biology, Academy of Sciences of the Czech Republic, Brno, Czech Republic

☯ These authors contributed equally to this work.
* se13@sanger.ac.uk

**Data availability statement:** Read data is available from the ENA at PRJEB76171. Reads

## Abstract

Many closely related species continue to hybridise after millions of generations of divergence. However, the extent to which current patterning in hybrid zones connects back to the speciation process remains unclear: does evidence for current multilocus barriers support the hypothesis of speciation due to multilocus divergence? We analyse whole-genome sequencing data to investigate the speciation history of the scarce swallowtails *Iphiclides podalirius* and *I. feisthamelii*, which abut at a narrow (∼25 km) contact zone north of the Pyrenees. We first quantify the heterogeneity of effective migration rate under a model of isolation with migration, using genomes sampled across the range to identify long-term barriers to gene flow. Secondly, we investigate the recent ancestry of individuals from the hybrid zone using genome polarisation and estimate the coupling coefficient under a model of a multilocus barrier. We infer a low rate of long-term gene flow from *I. feisthamelii* into *I. podalirius* - the direction of which matches the admixture across the hybrid zone - and complete reproductive isolation across ≈ 33% of the genome. Our contrast of recent and long-term gene flow shows that regions of low recent hybridisation are indeed enriched for long-term barriers which maintain divergence between these hybridising sister species. This finding paves the way for future analysis of the evolution of reproductive isolation along the speciation continuum.

## Author summary

Efforts to understand how new species evolve typically approach the problem through either: 1) investigating patterns of genetic exchange across 'hybrid zones' — where closely related species interbreed — or 2) modelling the demographic history of species divergence. Both approaches are capable of quantifying variation in genetic exchange, or 'gene flow', along the genome to identify regions of reproductive isolation; yet they

for sample IP 504 were generated by a previous study (https://doi.org/10.1093/g3journal/jkac193) and are available at the ENA at PRJEB51340. Input data for plots and statistics is available from https://github.com/samebdon/iphiclides_speciation_data. The bootstrapping was implemented using a custom script available at https://github.com/LohseLab/circular_bootstrap.

**Funding:** This work was supported by a European Research Council starting grant (ModelGenomLand 757648 to KL and DRL), an EastBio studentship from the Biotechnology and Biological Sciences Research Council including a stipend (to SE), a fellowship from the Natural Environment Research Council (NE/L011522/1 to KL), and a Ministerio de Ciencia e Innovación grant PID2022-139689NB-I00 (MICIU/ AEI/ 10.13039/501100011033 and ERDF, EU to RV) (https://erc.europa.eu/homepage, https://www.ukri.org/councils/bbsrc/, https://www.aei.gob.es). The funders had no role in study design, data collection and analysis, decision to publish, or preparation of the manuscript.

**Competing interests:** The authors have declared that no competing interests exist.

rely on different genetic signatures. While the former exploits allele frequency clines and patterns of linkage disequilibrium set up since the most recent range contact, the latter averages signatures over the history of divergence. Hence, we can contrast the genomic distribution of barriers acting on these different time scales to test how patterns of gene flow change across the speciation continuum. Here we use this strategy to capture the speciation dynamics of a pair of hybridising papilionid butterflies. Our results show that not only that these species continue to produce hybrids after more than a million years since the onset of divergence, but that there is a significant degree of concordance between patterns of gene flow observed along the genome across time scales.

## Introduction

A fundamental goal of speciation research is to understand the genetic basis of reproductive isolation (RI) between diverging species and quantify the demographic and selective processes that lead to a build-up of RI [1]. We now know that episodes of gene flow during speciation are not only possible [2–5] but frequent [6–10]: closely related species often continue to hybridise after millions of generations of divergence [11–14], yet remain distinct despite low levels of gene flow [15]. However, given the time scales over which speciation occurs, the processes that contribute to RI are likely to vary over time [16]. Stankowski and Ravinet [17] define the speciation continuum as a continuum of reproductive isolation, from incipience to complete hybrid inviability, and highlight that the species we observe today are at different stages along this continuum. For example, sister species in many temperate plant and animal taxa form secondary contact zones [18,19] which may be substantially younger than the onset of speciation. In other words, contemporary contact zones are likely one of many instances of secondary range contact generated by drastic environmental changes, such as glacial cycles over the last 800,000 years [20]. Hybrid zones (HZs) for such taxa are stable exactly because there is strong selection against admixed ancestry. Whilst it is evident that genetic divergence and differentiation varies along the genome [21,22], understanding the extent to which this reflects variation in the rate of gene flow requires explicit modelling of the interplay between migration, selection, genetic drift, and recombination.

Locally beneficial alleles may be selected against if they migrate into unfavourable environments and/or genetic backgrounds, reducing effective gene flow [23]. However, over time, gene flow may vary as a consequence of range shifts and/or other changes in demography caused by glacial cycles, and – as a consequence – the selective forces and targets underpinning RI may also vary over time. Periods of complete allopatry during which gene flow is interrupted facilitate the build-up of strong endogenous [24] or 'intrinsic' barriers [25]. Genetic studies of HZs show that recent introgression may vary considerably along the genome [26]: some genome regions harbouring strong incompatibilities act as strong barriers and show steep clines [27,28]. For example, steep clines in nature predicted [29] the approximate location of the second mammalian hybrid sterility gene Hstx2 years before its identification in the lab [30]. In other regions introgression may be unimpeded by selection or even advantageous [31–33]. Importantly, the barriers associated with RI since the onset of divergence may differ from those acting during secondary contact, many of which may have arisen recently [34,35] and may even have evolved to reinforce existing barriers in the face of gene flow [36]. To date, few studies have compared the targets of recent selection against heterospecific ancestry in hybrids with the architecture of long-term barriers to gene flow.

A recent meta-analysis of barriers to gene flow between the Western and Eastern European house mouse (*Mus musculus domesticus* vs *M. musculus musculus*) found no significant

overlap between postzygotic incompatibilities mapped in interspecific crosses in the lab and barriers to gene flow detected in the natural HZ of these taxa [37]. There are biological reasons for the non-repeatability between barriers detected in lab studies and HZs: firstly, the former are heavily biased towards barriers acting in the F1 or the first few backcross generations, the formations of which are extremely rare in natural HZs (which are typically several dispersal distances wide). Secondly, as argued by Frayer and Payseur [37] "the loci underlying barriers observed in the lab may be distinct from those that impede gene flow in nature" and may be involved in pre-mating isolation and/or extrinsic local adaptation. However, the lack of overlap found by Frayer and Payseur [37] may simply reflect a lack of statistical power resulting from the fact that their study pooled information from many clinal analyses with varying (but high) false positive rates and that mapping studies have low resolution in general; an $n$-generation lab cross study can only generate $O(n)$ crossovers per chromosome.

Thus, one may argue that contrasts of barriers acting since the most recent secondary contact of a species pair with barriers acting since the onset of divergence are not only more biologically relevant than contrasts with lab-based (early generation) hybrids, but also have greater statistical power, at least in principle. This simply reflects the fact that a typical HZ individual involves a much larger number of admixture generations, ancestors, and junctions in ancestry than any lab-cross.

Previous attempts to investigate the overlap of HZ introgression with long-term barriers relied on comparing outliers of genetic differentiation with cline-based analyses and yielded mixed results: Harrison and Larson [38] highlight several systems in which outliers of increased genetic differentiation show a reduction of recent introgression as measured by steeper genomic clines across secondary contact hybrid zones [28,39–46]. However, many other studies report mixed or negative results [40,47–54]. Interestingly, in some instances, fixed inversions have been shown to be associated with both 'islands of divergence' and steep clines [55–58]. While divergence outliers may correlate with regions of low HZ introgression, both the degree and significance of this association remain unclear.

In practice, comparisons between genomic clines and outliers of genetic differentiation have been limited for two reasons: firstly, summary statistics of the site frequency spectrum (such as $F_{ST}$) confound barrier effects with other population genetic processes and, by focusing on the most extreme outliers, limit sample size and power. Secondly, the power of clinal analyses is limited both by the need for extensive geographic sampling (which is impossible and/or prohibitively expensive for many natural HZ systems) and insufficiency of summary statistics; e.g. clines are often merely centered and scaled without allowing for variation in their shape and asymmetry.

The lack of comparisons of barriers to gene flow across time scales has been highlighted in a recent review on speciation genomics [35]. However, filling this gap requires quantitative inference frameworks that can distinguish barriers on both time scales from other evolutionary processes using the limited sample sizes available for most natural HZs. Such approaches have only become available recently.

## Speciation in *Iphiclides* butterflies

The southern European 'scarce swallowtails' *Iphiclides podalirius* and *I. feisthamelii* are large papilionid butterflies, typically associated with various species of *Prunus* and other Rosaceae bushes and trees. While *I. podalirius* ranges across the north of the Mediterranean from France to East Asia, *I. feisthamelii* is restricted to the Iberian peninsula and the northwest of Africa (Fig 1). Despite a long appreciation of the phenotypic differences between the two species [59], including genital morphology [60] and male ultraviolet wing patterns [61],

their taxonomic status has been disputed after DNA barcoding at the mitochondrial *COX1* locus revealed that the two taxa share mitochondrial haplotypes [62,63]. However, it has recently been suggested that mitochondrial genealogy reflects an introgression sweep most likely linked to infection by *Wolbachia* [61]. The two taxa diverged approximately 1.2 million years ago [20,64] and today, the species abut at a narrow (∼25 km) contact zone north of the Pyrenees [65] (Fig 1). While potential hybrids have been diagnosed based on morphology in a small set of museum specimens [66], the putative HZ has not yet been characterised genetically. Here, we quantify introgression across the HZ and estimate long-term effective migration rates ($m_e$) for this pair of sister species.

## Aims and objectives

We investigate the history of speciation and quantify both long and short-term barriers to gene flow between *I. podalirius* and *I. feisthamelii*. We use two recently developed minimal-assumption inference methods `gIMble` [67] and `diem` [68] on whole-genome sequencing (WGS) data to quantify long-term barriers and selection against recent introgression in the HZ, respectively. Firstly, we infer variation in the long-term rate of effective migration along the genome since species divergence under an explicit demographic model and locate putative barriers to gene flow with `gIMble`. Secondly, we use the genome polarisation framework `diem` to quantify recent barriers to introgression in six putative hybrid individuals sampled from the *Iphiclides* HZ (Fig 1) and characterize the multilocus barrier to ongoing hybridisation between these species. Finally, we investigate the overlap between long-term barriers to gene flow and genomic regions that are depleted for recent introgression. We address the following questions:

1. What is the direction and rate of gene flow between *I. podalirius* and *I. feisthamelii* since their initial divergence?
2. What fraction of the *Iphiclides* genome acts as a long-term barrier to gene flow, and what are the properties of barrier regions?
3. What is the evidence for recent gene flow between the species across the HZ?
4. Is introgression across the HZ impeded by long-term barrier regions, i.e. do the barriers acting over these two different timescales overlap?
5. What is the strength of the multilocus barrier acting against introgression in the HZ?

## Results

### Interspecific variation is consistent with hybridisation in *Iphiclides*

We generated WGS data for six individuals from the *Iphiclides* hybrid zone (we will refer to these as "the HZ set") in southern France (Fig 1B) and 14 individuals sampled throughout the ranges of both species ("the non-HZ set", Fig 1A). The non-HZ set includes six samples of *I. podalirius* and eight of *I. feisthamelii*.

Visualising genetic variation in a PCA reveals distinct clusters both within and between species. The first principle component (PC1) captures interspecies differentiation between *I. podalirius* and *I. feisthamelii* (≈ 23% of the variation, Fig 2). Samples assigned morphologically to each species form two clusters along PC1 with individuals from the HZ (Fig 2) falling between the two parental species clusters (Fig 2). The two North African *I. feisthamelii* samples are separated from the European *I. feisthamelii* samples along PC2 (≈ 14% of the variation, Fig 2). Genetic diversity is slightly larger in *I. feisthamelii* compared to *I. podalirius* (Table 1.).

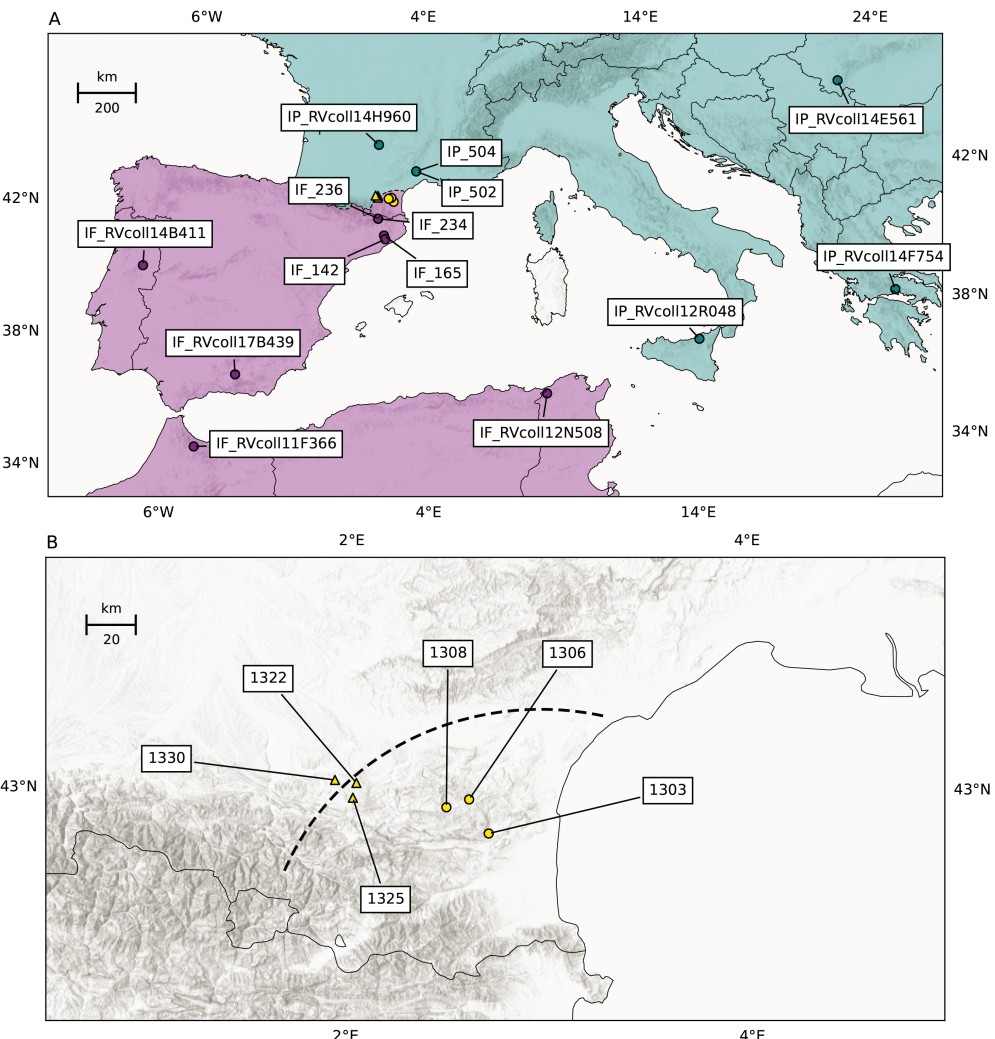

**Fig 1. (A) Sampling locations and ranges of *I. feisthamelii* (purple) and *I. podalirius* (teal) butterflies.** The samples collected from the hybrid zone (HZ) are shown in yellow. (B) Sampling locations of butterflies from the *Iphiclides* HZ. The dashed line represents the approximate HZ center, based on samples collected by Lafranchis et al. [66]. The circular samples resemble *I. feisthamelii* (HI < 0.1), the triangular samples are intermediate hybrids (HI > 0.1). Maps were generated using the Python 'basemap' package with Natural Earth's 10m country and coastline datasets (available here), and a relief layer from the ArcGIS Rest Service [90]. The basemap plotting code is available here.

We identified runs of homozygosity (ROH) > 100kb using PLINK (v1.9) [69] to estimate inbreeding via $F_{ROH}$. One *I. podalirius* sample from Sicily (RVcoll12R048) was particularly inbred ($F_{ROH} \approx 0.25$), and two other *I. podalirius* individuals, one from Romania (RVcoll14E561) and one from the HZ (1325) were somewhat inbred ($F_{ROH} \approx 0.076$ and 0.061 respectively). $F_{ROH}$ in the remaining samples (including all *I. feisthamelii*) was negligible (S1 Table).

## Evidence for post-divergence gene flow from *I. feisthamelii* into *I. podalirius*

We use the composite-likelihood method `gIMble` [67] to infer long-term barriers to gene flow. Although the speciation history of *Iphiclides* most likely involved glacial cycles of isolation and secondary contact, our aim is not to reconstruct this likely complex and

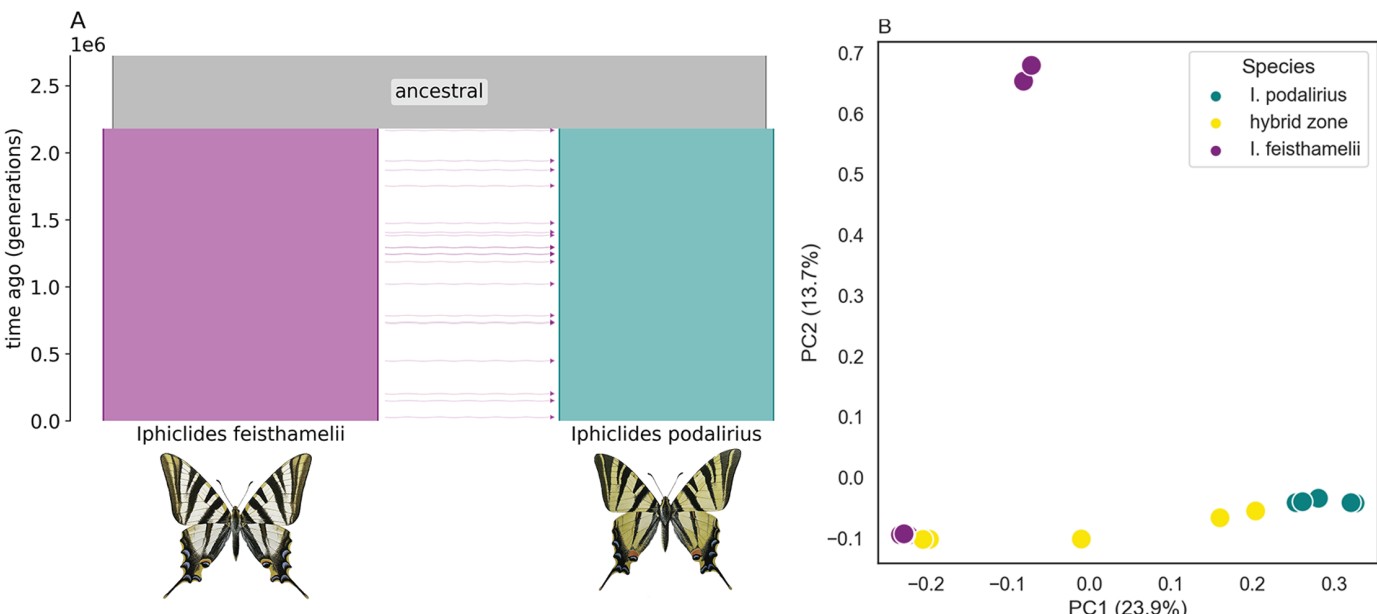

**Fig 2. (A) The background demographic history of species divergence and gene flow, the height and width of populations, is relative to the maximum likelihood estimates under the** $IM_{2,\rightarrow pod}$ **model (Table 2.).** This figure was generated using *demes* [91]. (B) PCA of *Iphiclides* sampled across Europe; *I. feisthamelii* and *I. podalirius* samples are shown in purple and teal respectively. Samples from the HZ are shown in yellow: PC1 captures differences between the two taxa, PC2 geographic structure, particularly the separation between North African and European *I. feisthamelii*.

**Table 1. Estimates of genetic diversity (H, $\pi$, and $\theta$), divergence ($d_{xy}$), and differentiation ($F_{ST}$) at intergenic (I) and fourfold degenerate (4D) sites between *I. feisthamelii* and *I. podalirius*.** Estimates for 4D sites are taken from [20].

|  | $H_I$ | $H_{4D}$ | $\pi_I$ | $\theta_I$ | $d_{xy,I}$ | $d_{xy,4D}$ | $F_{ST,I}$ | $F_{ST,4D}$ |
|---|---|---|---|---|---|---|---|---|
| *I. feisthamelii* | 0.00676 | 0.00794 | 0.00682 | 0.00692 | 0.0244 | 0.0275 | 0.594 | 0.575 |
| *I. podalirius* | 0.00585 | 0.00521 | 0.00652 | 0.00659 |  |  |  |  |

dynamic demographic history in any detail, but rather to capture the variation in long-term gene flow along the genome with the fewest number of parameters. We therefore fit an *IM* model that assumes a constant rate of $m_e$ through time but allows for heterogeneity in $m_e$ along the genome [9,67]. Contrasting the support for a background *IM* model and a history without gene flow in genomic windows gives a measure for the cumulative local strength of barrier loci. Since this analysis of barriers relies on the assumption of a two-population history, HZ and North African samples were excluded. We summarise genetic variation in short 64 base 'blocks', which are assumed to be non-recombining, under no direct selection, and to evolve with a constant mutation rate. These assumptions allow modelling the shared genealogical history of closely linked variants in the composite likelihood framework implemented in `gIMble`. To maximise the density of neutrally evolving sites, we follow Laetsch et al. [67] and restrict this analysis to intergenic sequence which have similar per site diversity and divergence to fourfold degenerate (4D) sites in coding regions (Table 1). After applying coverage-based filters (see Methods), our analyses include 39% of the genome ($\approx$ 160 out of 408 Mb).

We fit a series of demographic models to the blockwise site frequency spectrum (bSFS) of the whole genome: a model of strict divergence (*DIV*) and an *IM* model with migration in

either direction with three $N_e$ parameters ($IM_{3,\to pod}$ and $IM_{3,\to fei}$). Out of these three scenarios (Table 2), the best fitting history is an $IM_3$ model with unidirectional gene flow (forwards in time) from *I. feisthamelii* into *I. podalirius* ($IM_{3,\to pod}$). Since an $IM$ model will always fit at least as well as the (nested) $DIV$ model, we compared the observed improvement in model fit (ln $CL$) relative to the null distribution of ln $CL$ which we obtained from simulating 100 data sets under the best fitting $DIV$ history (see methods). This parametric bootstrap confirms that the $IM_{3,\to pod}$ model does indeed fit significantly better than a $DIV_3$ history (S1 Fig).

Under the background/global $IM$ model, we infer a split time ($T$) 2,180,000 generations ago (Fig 2 and Table 2) and $N_e$ estimates of 483,000, 377,000 and 1,140,000 for *I. feisthamelii* and *I. podalirius* and the ancestral population, respectively. This split time is close to previous estimates for the species pair [20]. The long-term genome-wide rate of gene flow $m_e$ from *I. feisthamelii* into *I. podalirius* is $4.73 \times 10^{-8}$ which corresponds to $M/2 = 2N_e m_e = 0.046$ migrants per generation.

## Extensive genome-wide reproductive isolation between *I. feisthamelii* and *I. podalirius*

We infer the effective migration rate $m_e$ along the genome in sliding windows each composed of 28,125 intergenic blocks (which corresponds to a median window span $\approx$150 kb). For each window, we estimate parameters under the best fitting $IM_{3,\to pod}$ history using a pre-computed grid. Note that whilst the split time $T$ is fixed globally, the remaining four parameters ($N_e$ for each population and $m_e$) are estimated locally, i.e. per window. The estimates of local $N_e$ are approximately normally distributed (S2 Fig). In contrast, local $m_e$ estimates have a strongly leptokurtic distribution with a peak at $m_e = 0$ and a long tail up to the maximum value in the grid (S2 Fig). Note that the large number of windows in the largest $m_e$ bin reflects the fact that our grid of $m_e$ estimates necessarily truncates the $m_e$ distribution, but does not affect our analyses of barriers.

Following Laetsch et al. [67] we label windows as barriers to gene flow if a $DIV$ history ($m_e = 0$) has greater marginal support than an $IM$ history assuming the best fitting genome-wide value of $m_e$ (Table 2) and this difference in marginal support is supported by a parametric bootstrap (see methods), i.e. $\Delta_{B,0} > 0$. Applying this strictest possible barrier definition, we find that barriers to gene flow are widespread in the *Iphiclides* genome, making up 20% of windows across all chromosomes. Combining overlapping barrier windows, we define 555 barrier regions across all autosomes that cover 33% of the genome ($\approx$ 143Mb). The average length of barrier regions is $\approx$ 257 kb (median $\approx$ 200 kb) with a maximum of 1.4 Mb.

## The genomic correlates of barrier regions

Assuming that the causal loci underlying barriers are in or near genes, we predict an enrichment of long-term barrier regions for coding sequence. This pattern has been observed in *Heliconius melpomene* and *H. cydno* [67] as well as scans for pairwise incompatibilities in natural hybrid populations of swordtail fish [70]. Contrary to this expectation, we find that both

**Table 2. Maximum composite likelihood parameters for three demographic models of species divergence.** We use the $IM_{3,\to pod}$ model for barrier inference. $\Delta$ ln $CL$ is relative to the best supported model (bold).

| model | $N_{fei}$ | $N_{pod}$ | $N_{anc}$ | $T$ | $m$ | $\Delta$ ln $CL$ |
|---|---|---|---|---|---|---|
| *DIV* | 459,000 | 416,000 | 1,240,000 | 1,810,000 | - | -389,374 |
| **$IM_{3,\to pod}$** | **483,000** | **377,000** | **1,150,000** | **2,180,000** | **4.73E-08** | **0** |
| *$IM_{3,\to fei}$* | 438,000 | 429,000 | 1,190,000 | 2,020,000 | 2.80E-08 | -238,736 |

the density of coding sequence (CDS) and repeats are reduced in barrier regions (S3 Fig, see discussion). Given that our windows are both overlapping and genetic variation is autocorrelated along the genome (due to LD and various other sources of autocorrelation), standard statistical methods that assume independence are inappropriate for assessing the significance of differences between barrier and non-barrier windows and would lead to vastly overconfident conclusions. Instead, we use a circular data bootstrap approach to quantify the statistical support for correlates of barriers. This is inspired by a similar circular resampling scheme used by Yassin et al. [71] and Nouhaud et al. [72] who resampled by circularising the genome and sampling datasets with fixed offsets. We modify this procedure to sample with random offsets and circularising each chromosome (rather than the whole genome). This allows us to obtain null distributions for any genomic measure (e.g. CDS density) for random sets of genomic windows that have the same distribution between and distances/clustering within chromosomes as the set of barrier windows inferred by $gIMble$. Applying this data bootstrap confirms that both CDS and repeat density are indeed significantly reduced in long-term barriers compared to non-barrier windows (S3 Fig).

Models of local adaptation under migration-selection balance predict a concentration of barriers in regions of low recombination and a positive correlation between $m_e$ and recombination [73]. While we lack direct estimates of recombination for *Iphiclides*, we can use chromosome location as an indirect proxy for recombination over two scales. First, given the requirement of a single cross-over per meiosis [74], chromosome length is inversely related to recombination rate [75–77]. Second, an ubiquitous feature of meiosis in Lepidoptera is that – despite the lack of centromeres – recombination is reduced in the centre of chromosomes. Indeed, heterozygosity is strongly correlated with chromosome length in both *I. podalirius* (Pearson's $\rho$ = -0.827, p = 3.18e-9) and *I. feisthamelii* (Pearson's $\rho$ = -0.856, p = 3.19e-9) and is lower towards the center of chromosomes (S4C Fig and S5 Fig). We therefore expect both long chromosomes and chromosome centres to be enriched for barriers to gene flow. In line with these predictions, we find that barriers are correlated with recombination rate variation between and within chromosomes. First, both the proportion of barrier sequence (Pearson's $\rho$ = 0.43, p = 0.0241, S4A Fig) and the average $m_e$ (Pearson's $\rho$ = -0.55, p = 0.00195) are correlated with chromosome length. Second, we observe that, on average, barrier regions are closer to the centre of chromosomes than non-barrier regions (circular bootstrap, p = 0.001 S4C Fig). The number of barrier regions varies widely between chromosomes: e.g. chromosomes 29 and 31 harbour no and one barrier region, respectively. In contrast, chromosomes 19 and 17 have 53 (29.3% of windows) and 49 (29.5% of windows) barrier regions, respectively (S6 Fig).

## Genome polarisation reveals complex hybrids

Chromosome-painting of individual genomes via diagnostic markers provides a direct way to visualize the mosaic ancestry of HZ samples [78–80]. While many studies have relied on assigning hybrid genotypes based on a reference panel assumed to reflect 'pure' ancestry, this assumption is both unnecessary and biases inference against introgression [81]. We labeled the ancestry of alleles at all non-singleton SNPs with respect to the focal (i.e. between species) barrier using the EM algorithm implemented in $diem$ [68]. This approach requires no *a priori* defined sets of reference individuals or candidate variants and assigns genotypes of 0/0, 1/1 or 0/1 at each variant position which correspond to homozygous for each species or heterozygous by source respectively. Note that although all samples were included in the polarisation, we focus our analysis of recent introgression on the "HZ set" of six individuals (Fig 1B).

Genome polarisation shows that the HZ individuals (Fig 1B) have an alternating pattern of *podalirius—feisthamelii* ancestry along all autosomes (Fig 3), which can only be generated by hybridisation after divergence. We find substantial heterogeneity in the degree of hybridisation, as quantified by the hybrid index (*HI*) – defined as the average across genotype assignment of an individual – both between individuals and chromosomes (Fig 4). The two most intermediate HZ samples, 1325 (*HI* = 0.712) and 1322 (*HI* = 0.420), show large stretches of all three possible ancestries, a pattern that reflects a complex history of hybridisation: the expected *HI* of a simple *n*-th generation backcross hybrid is $(1/2)^n$ with homozygous tracts for one species only. While we expect substantial variation around this expectation, successive generations of simple backcrosses move the expected *HI* towards the parental values of 0 or 1 along the sides of the ternary plot of *HI* against heterozygosity (*H*). In other words, interspecific *H* (the proportion of genotypes with an allele from each side of the barrier) remains maximal. Thus, the relationship between *HI* and interspecific *H* shows that samples 1325 and 1330 are complex backcrosses (i.e. their ancestry is incompatible with successive backcrosses with pure parental individuals), and suggests that in both cases introgression is towards *I. podalirius* (Fig 4A). Interestingly, all HZ samples except 1322 show a deficit in interspecific heterozygosity relative to Hardy-Weinberg equilibrium (Fig 4). While these ternary diagrams show genotype samples from genomes rather than from loci, Hardy-Weinberg equilibrium remains the null expectation for localities at equilibrium for an admixture process uniform along the genome. Thus, the deficit in heterozygosity suggests local substructure/inbreeding in the HZ and/or heterogeneous selection against heterozygous (by source) ancestry, which may be driven by partially dominant incompatibilities. Strikingly, we find that the six HZ individuals (which include two male samples) only exhibit one Z chromosome of mixed ancestry (1325, a female), which suggests that the Z is a substantially greater barrier than autosomes of equivalent length, as predicted by numerous theoretical models [82–85].

## Long-term barriers to gene flow are associated with regions of low hybridity

Setting up a comparison between long-term barriers to gene flow and introgression across the *Iphiclides* HZ requires a measure of barrier strength that extracts information contained in the small sample of HZ individuals. Classic theory for hybrid zone barriers is couched in terms of cline parameters which cannot be meaningfully estimated from a sample of six individuals. Nevertheless, it is clear that chromosome paintings even for a small sample of individuals contain a wealth of information about recent introgression. For example, assuming that our sample of six HZ individuals is centered, a genomic region without any gene flow (i.e. a complete short-term barrier region) is expected to be painted entirely homozygous for *I. podalirius* and *I. feisthamelii* ancestry for three individuals on either side of the centre (clines would be maximally narrow, stepped, and steep). To measure the local strength of barriers in a small set of polarized hybrid genomes, we estimated $\bar{D}$ [86], the multi-site mean pairwise LD, in windows defined in the `gIMble` analysis. This is numerically equivalent to the variance in *HI* and captures the strong LD seen along stretches of co-introgressing variation: when minimal for polarised data (*D* = 0), states of sites are uncorrelated, and when maximal (*D* = 0.25), each sample reflects pure ancestry.

We find that — at the scale of `gIMble` windows ($\approx$ 100 kb) — $\bar{D}$ is negatively correlated with $m_{e,i}$ (Pearson's $\rho$ = -0.57, p = 0.0). To test whether short-term barrier strength (as measured by $\bar{D}$) is greater within long-term barrier windows than expected by chance, we conducted two different circularised bootstrap schemes to obtain null distributions for $\bar{D}$:

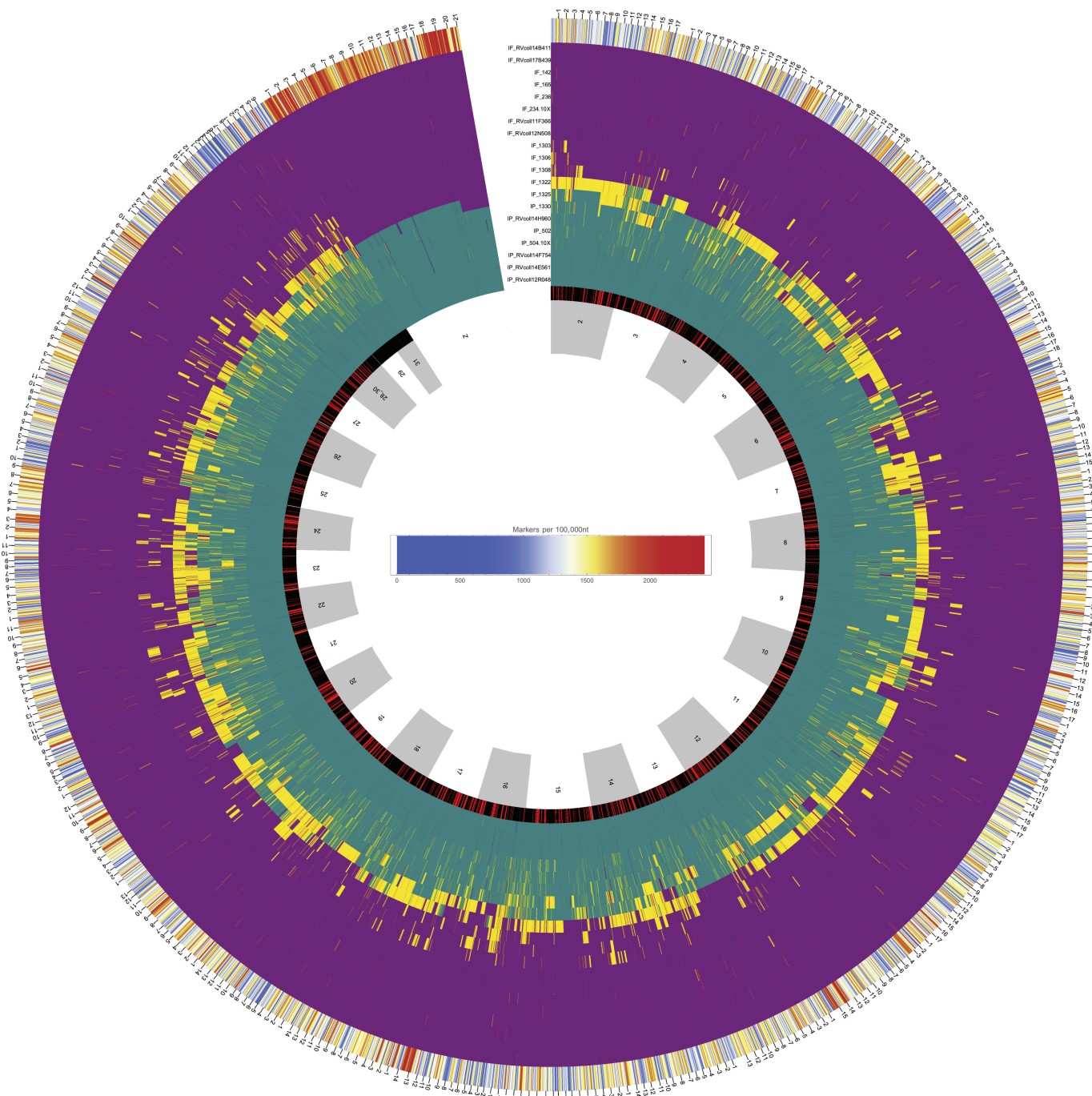

**Fig 3. Circular representation of the location of barriers identified using `gIMble` and the polarity of diagnostic markers for each sample across linkage groups.** The inner ring shows the location of each chromosome in alternating grey and white. Moving outwards, the next ring indicates the location of barriers to gene flow (in red) and non-barrier/migrating regions (in black). The remaining rings show the genotype of each sample at each diagnostic marker. Teal bars are diagnostic of *I. podalirius*, purple bars are diagnostic of *I. feisthamelii*, and yellow bars are heterozygous for markers diagnostic of each species. The outermost ring shows the density of high DI sites. The location of each megabase of sequence for each chromosome is indicated on the outside of the circle.

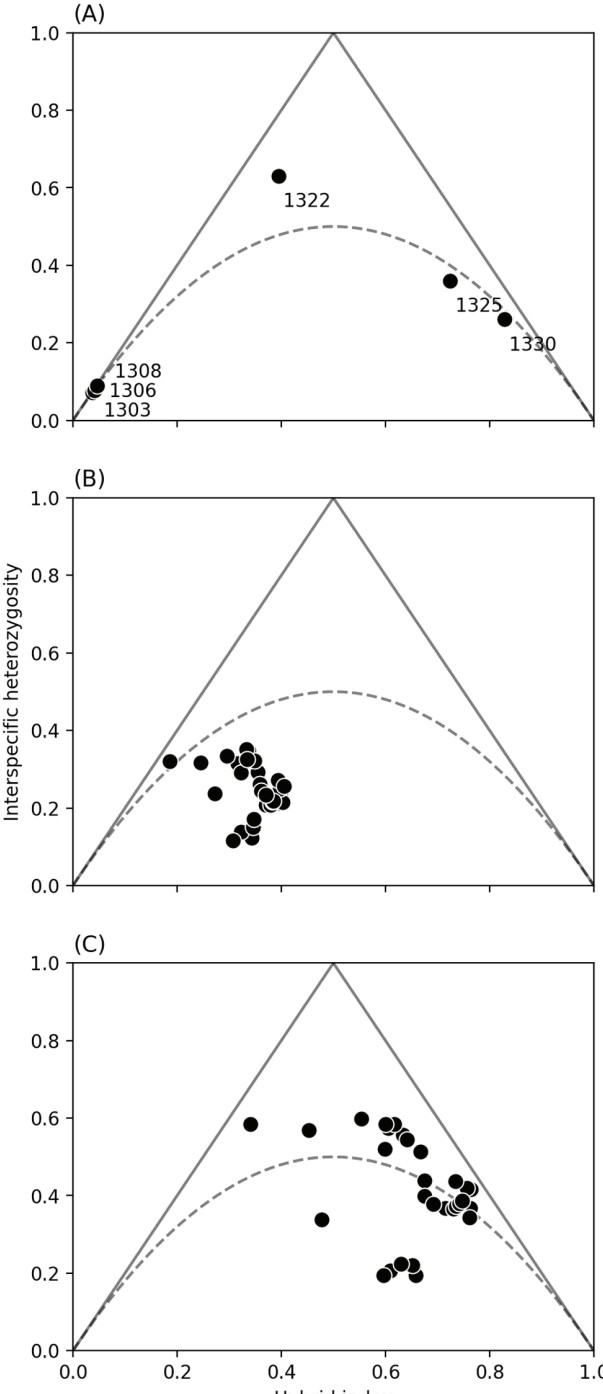

**Fig 4. Hybrid index (*HI*) versus interspecific heterozygosity (*H*) for samples collected from the *Iphiclides* hybrid zone (HZ).** (A) Mean values for each HZ sample. (B) Mean values for each chromosome including all samples from the HZ. (C) Mean values for each chromosome excluding the three *I. feisthamelii*-like samples (1303, 1306 and 1308, $HI \approx 0$). The dashed line indicates the expectation under Hardy-Weinberg equilibrium.

either circularising each chromosome (accounting for differences in the number of barriers between chromosomes) or circularising the whole genome (not accounting for chromosome effects). We find that the mean $\bar{D}$ for barrier windows is significantly greater than either of these resampled distributions (circular bootstrap, $p<0.001$, Fig 5B). This suggests ongoing selection against hybrid ancestry within the HZ at long-term barrier loci (Fig 5A). While the upward shift in the resampled distributions when accounting for chromosome of origin suggests that the between-chromosome variation in recombination rate does contribute to the overlap between long and short-term barriers, this overlap cannot be explained by the fact that both $\bar{D}$ (Pearson's $\rho = 0.74$, p = 4.84e-06) and barrier density are correlated with chromosome length (S4A Fig and S4B Fig).

While $\bar{D}$ is an obvious measure of short-term barrier strength that is straightforward to compute from a set of `diem` polarized genomes, it is important to consider alternatives. A potential measure of the short-term barrier effects in a set of hybrid samples is the density of unique ancestry junctions. The number of unique ancestry junctions in a genomic region is negatively correlated with barrier strength [87], as selection against heterozygous ancestry limits the decay of admixture tracts generated by successive introgression events. Thus, we expect barrier regions to be populated by fewer and larger blocks of co-ancestry, and necessarily, fewer junctions than migrating regions. Consistent with this prediction, we find that barrier regions contain almost half the number of ancestry junctions than expected by chance (circular bootstrap, $p<0.001$, Fig 5C and 5D).

However, both measures of short-term barrier strength rely on markers with high diagnostic indices which occur at higher density in `gIMble` barrier windows compared to random windows (circular bootstrap, $p<0.001$, Fig 5E and 5F). Given that the local density of high DI markers must be a result of barriers acting over a range of timescales, it is crucial to test whether the overlap of long and short-term barriers we find (in the form of an excess of $\bar{D}$ or reduction of junctions in `gIMble` barriers) simply reflects the density of high DI markers. To assess the robustness of our findings to heterogeneity in high DI marker density, we repeated our comparison between barrier and migrating windows (in terms of $\bar{D}$ and junction density) for a dataset in which high DI markers were downsampled to be uniformly distributed (see methods). We find that our core result of reduced introgression across the six HZ samples in long-term barriers holds equally when we remove the heterogeneity of high DI markers (S8 Fig and S9 Fig). Our results are similarly robust to the exclusion of individual HZ samples (circular bootstrap, $p<0.001$ in all six instances for each metric).

## Modelling the history of introgressed blocks in complex hybrids

We have so far interrogated the data for the six HZ individuals in terms of the variation in the strength of barriers acting since the most recent secondary contact, both between and within chromosomes. However, theoretical models of multilocus barriers consider the aggregate effects of many loci under selection. This creates a genome-wide barrier that (assuming a model of uniformly distributed selection targets) can be captured by the ratio between the selection pressure and the recombination rate, a.k.a. the coupling coefficient $\theta = S/R$ [88,89]. Thus an obvious question is: what strength of multilocus selection is the mosaic ancestry of the six HZ individuals compatible with? To address this, we considered the length distribution of introgression tracts. If we make the simplifying assumption that introgression across the HZ into either species started at time $T$, occurs at a constant rate $m$, and only involves backcrosses (i.e. the recipient population is infinitely large), the lengths of admixture tracts depend only on the coupling coefficient $\theta$ [88,89]. More precisely, the equilibrium solution for the gradient of the admixture tract length distribution on a log-log scale is $-\frac{3+\theta}{1+\theta}$ (see Fig 6,

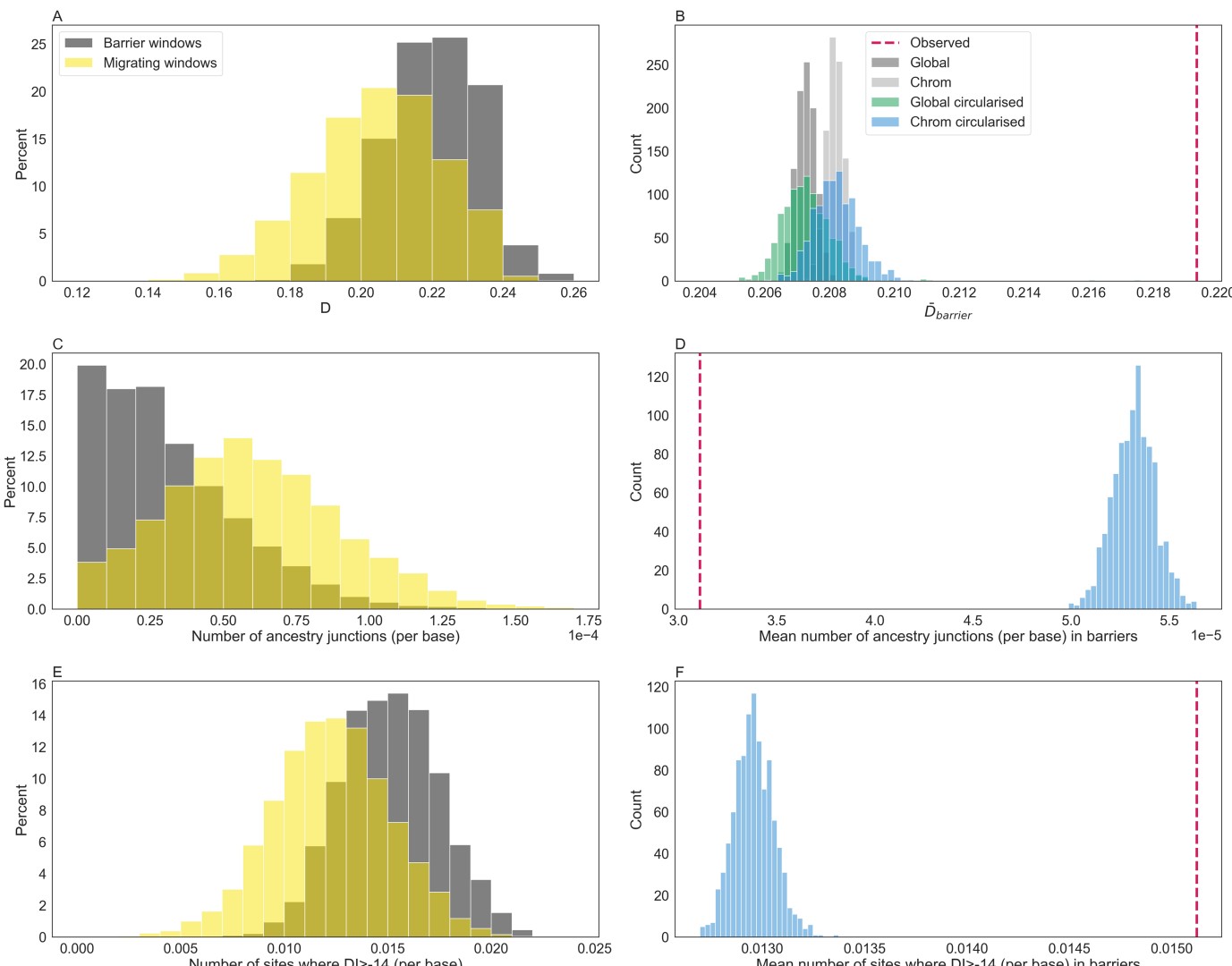

**Fig 5. The distributions of $\bar{D}$, the number of unique ancestry junctions, and the number of strongly diagnostic sites (A/C/E) across `gIMble` barrier windows (grey) and non-barrier/migrating windows (yellow) and their corresponding bootstrap results (B/D/F).** The latter two metrics have been corrected for window span. Both $\bar{D}$ calculated per window (A and B, circular bootstrap, $p<0.001$) and the number of highly diagnostic markers (E and F, circular bootstrap, $p<0.001$) is greater within long-term barriers than in non-barrier windows. The number of unique ancestry junctions is lower within long-term barriers than in non-barrier windows (C and D, circular bootstrap, $p<0.001$). Null distributions of $\bar{D}$ were generated using four different resampling schemes (B). In each instance, random values of $\bar{D}$ were drawn without replacement from the empirical distribution of window-wise $\bar{D}$ to generate datasets corresponding to the number of barrier windows. We repeat each resampling 1,000 times and compare the distribution of mean $\bar{D}$ to the observed value. Firstly, we sample datasets from the entire genome with (green) and without (black) circularising (see methods). Secondly, we resample per-chromosome accounting for differences in the number of outliers between chromosomes, also with (blue) and without (grey) circularising. We only show the most conservative test - the circular bootstrap accounting for chromosome-of-origin - for the number of junctions (D) and the number of strongly diagnostic sites (F).

S1 Appendix). Thus, to learn about the direction and dynamics of recent gene flow across the HZ, we fit the distribution of admixture tract length of colinear, introgressed ancestry to the analytic expectation under this model [89]. In the absence of a recombination map, we measured the length of admixture tracts ($x$) relative to chromosome length, i.e. we assumed that each autosome has a map length of 25 cM, which corresponds to an average of one cross-over

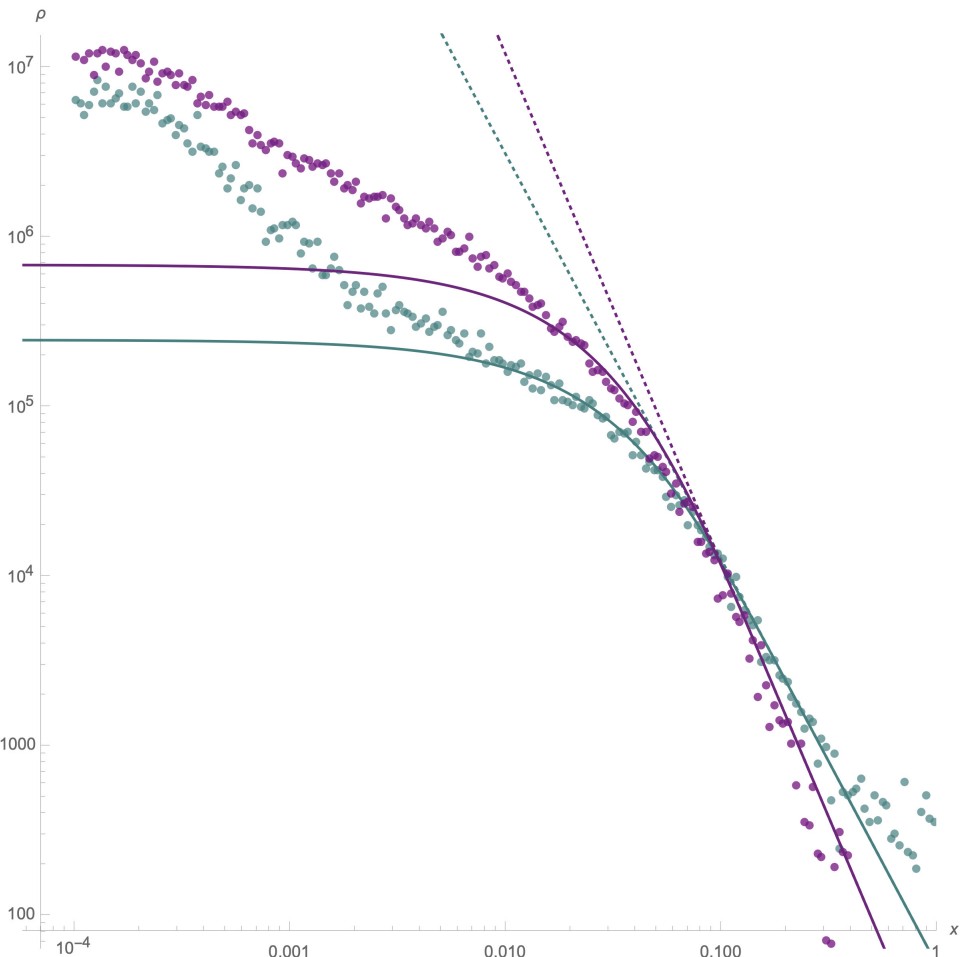

**Fig 6. Distribution of sizes of (purple points) _I. feisthamelii_ tracts introgressing into _I. podalirius_, and (teal points) _I. podalirius_ tracts into _I. feisthamelii_, on a log-log scale.** Note, as most introgression is heterozygous, the introgressing tracts largely correspond to the yellow tracts in Fig 3. These are compared to theoretical predictions (solid lines) for exchange between two infinite demes (S1 Appendix A-5). Introgression into _I. podalirius_ is plotted for [_M, S, R, T_= 15, 0.11, 0.25, 150]. Introgression into _I. feisthamelii_ is plotted for [_M, S, R, T_ = 1.5, 0.0, 0.25, 275]. The difference in gradients on the right indicates stronger coupling (_S/R_) of the _I. podalirius_ background despite more migrants _M_ per generation. The distribution of small blocks towards the left does not match theory (see Results), making the time-since-contact estimates _T_ lower bounds only.

event per male meiosis (female butterflies are achiasmatic). To avoid block lengths being fragmented by rare errors and gene conversion events, we kernel smoothed `diem` labeling along chromosomes at a scale of $10^{-4}\times$ chromosome length, which corresponds to >1kb for most autosomes.

Considering long tracts (_l_>0.04), neutrality provides a good fit for the distribution of tracts of _I. feisthamelii_ ancestry in _I. podalirius_ (i.e. a gradient of -3). In contrast, the size distribution of _I. podalirius_ admixture tracts in _I. feisthamelii_ is best explained by θ = 0.44 (Fig 6). This asymmetry mirrors the asymmetry of long-term gene flow under the _IM_ history inferred by `gIMble` and suggests that gene flow has been stable for much of the last 150 - 275 generations (75 - 140 years). In contrast, over the same timescale _I. podalirius_ admixture into _I. feisthamelii_ has been strongly selected against. Interestingly, we find that the admixture

tract length distribution in neither direction involves a simple asymptote for short tracts as predicted for a secondary contact initiated at time $T$ with constant $m$. This may be due to several factors: first, gene flow in the more distant past may have been genuinely lower, which — given the dependence of *Iphiclides* on orchards and other anthropogenic habitats — may reflect changes in landscape use. Secondly, our simplistic measure of $x$ as a proportion of chromosome length is inadequate for short blocks; given that recombination is known to vary substantially along butterfly chromosomes [76]. Thirdly, our model is unrealistically simplistic in that it ignores space: it assumes introgression into an infinitely large panmictic population. In a spatially continuous population, we expect shorter tracts with increasing distance from the HZ, which is indeed what we observe in *Iphiclides*. Fourthly, violating our infinite population size assumption, hybrids in finite populations can interact, producing complex admixture tracts. This may inflate the distribution of small tracts in both admixture directions and explain the asymmetric deviation from expectation we see between fits. Finally, it is also plausible that the polygenic model of barrier architecture assumed by Baird [89] breaks down over short physical scales, because it does not include the possibility that increasingly short blocks may have increasingly variable fitness effects (including the possibility of adaptive introgression).

## Discussion

Much of the recent research on the genomics of speciation has focused either on fitting demographic models to estimate migration over timescales of $N_e$ generations [92–95] or on investigating barriers to recent introgression (10s or 100s of generations) in studies of HZs (e.g. [29, 52,96,97]), natural hybrid populations [70] and laboratory crosses (e.g. [98,99]). However, there are surprisingly few attempts to understand how barriers to gene flow over different timescales are related. Here we have fitted explicit demographic models of speciation to infer the heterogeneity in long-term effective migration between two sister species of *Iphiclides* butterflies. We intersect this inference with signals of recent, heterogenous introgression across a HZ estimated from a small set of HZ individuals using genome polarisation.

### Evidence for both long-term and recent gene flow between scarce swallowtail sister species

We find that the long-term demographic history of *I. podalirius* and *I. feisthamelii* is well approximated by an IM model with a low rate of unidirectional migration ($M = 2N_e m = 0.046$) from *I. feisthamelii* into *I. podalirius*. Thus our analysis adds to the growing number of young species pairs for which a signal of long-term and ongoing migration has been identified, including great apes [100], butterflies [8,101,102], mollusks [103], angiosperms [104], birds [105], *Drosophila* [106], and many more.

   However, we note that the global effective migration rate ($m_e$) we infer in *Iphiclides* is considerably lower than estimates obtained for sister species pairs of *Heliconius* [67] and *Brenthis* [102] butterflies, both of which are of comparable age (S7 Fig). Although this low background level of genome-wide migration reduces the power to identify individual barrier regions, it still allows quantification of the variation of $m_e$ and its correlates along the genome. Furthermore, we find evidence for ongoing introgression between the two *Iphiclides* species across a HZ in the form of complex hybrids. Relating the distribution of admixture tract lengths to analytic expectations under a model of secondary contact [89] suggests that recent gene flow into *I. podalirius* is neutral, while gene flow into *I. feisthamelii* is strongly selected against. This asymmetry is concordant with the inferred direction of long-term gene

flow. It is also consistent with genetic load arguments, which predict stronger selection against admixture tracts derived from the taxon with lower $N_e$ [107], i.e. *I. podalirius* in this case.

More generally, it is encouraging how much information about both recent and long-term introgression is contained in a small sample of individual genomes. Given that most species pairs do not have HZs, such long-term barrier inference is the only genomic information available about their speciation history. Even when HZs exist, they may not be amenable to clinal analyses due to sampling constraints as is the case for *Iphiclides*. However, our analyses demonstrate that the genome-wide barrier, which is described by the coupling coefficient and the recent history of gene flow can be estimated from a handful of HZ individuals. It is perhaps surprising how well the admixture tract lengths we observe in *Iphiclides* fit analytic expectations under the simplest possible model of secondary contact which assumes panmixia, infinite population size and weak selection [89]. The fact that the observed length distribution deviates from this expectation for short tracts suggests that small samples of hybrid genomes contain additional information about admixture. In particular, it would be interesting to fit models of more realistic barrier architectures that assume a large but finite number of loci. Furthermore, classic theory on Fisher junctions for HZs only considers the decay of admixture tracts due to crossover events [88]. However, the internal decay of admixture tracts due to gene conversion events has so far been ignored and overlays the effects of an additional clock.

The ability to estimate the genomic distribution of block sizes (Fig 6) and relate it to theory [89] developed long before genomic data were available is thrilling. The distribution of large blocks is expected to reach equilibrium quickly, and appears log-log linear, suggesting relatively strong coupling of 0.44 at least in one direction. Neither direction reaches the 'tipping point' coupling of $S/R = 1$ [88], where multilocus clines 'congeal', however, that sharp threshold at equilibrium is misleading. Because small blocks equilibrate very slowly, it would take longer than an interglacial period for a sharp distinction in secondary contact outcome to be perceivable [89] (Fig 2).

## Long and short-term barriers to gene flow

The fact that barriers, inferred over these two time-scales of evolution, overlap shows that a significant subset of barriers is persistent and acts at very different points of the speciation continuum. This suggests that regions of the genome that maintain reproductive isolation between species in the long-term are also relevant early on in species divergence. Indeed, in *Heliconius* butterflies it has been shown that the same wing pattern genes maintain species differences both across HZs [108] and in deep time [67]. However, in contrast, the congruence in barrier landscapes across timescales we find in *Iphiclides* is not restricted to a small number of large effect genes, but rather a genome-wide phenomenon, suggesting a polygenic barrier architecture.

We would argue that the internal comparison of long and short-term barrier landscapes we have conducted here is a more promising avenue for testing of models of speciation than comparisons/contrasts of species pairs at different stages, which invariably differ in speciation history and barrier architecture [17,109,110]. Thus, an important direction for future work is to develop quantitative predictions for the temporal change in barrier landscapes. Under an allopatric null model, which may apply to *Iphiclides* during glacial periods, pairwise intrinsic incompatibilities are assumed to accumulate at random positions in the genome and genome-wide coupling is expected to increase quadratically with time [111]. Under this model, we expect a limited amount of temporal barrier overlap which arises from the fact that barrier loci that establish early have a greater effect on long-term $m_e$ than later barriers. In contrast,

verbal models of 'divergence hitchhiking' assume that early barriers expand locally [4,112], and so may predict a greater degree of temporal overlap. While theoretical models show that locally expanding barriers are possible — given sufficiently strong selection and linkage [113] — there is so far not much empirical evidence that locally expanding 'islands of speciation' are a common feature of speciation.

### The architecture of barrier loci

We found substantial variation both in the size of barrier regions and in the proportion of barrier sequence per chromosome. Specifically, barriers to gene flow aggregate on large chromosomes and towards chromosome centres. Both are regions of the genome where recombination rates are reduced. Thus, it appears on a broad scale that the architecture of reproductive isolation between *Iphiclides* species is strongly linked to recombination rate heterogeneity, as would be expected from barrier loci that individually confer small effects. This is consistent with previous research on *Heliconius* butterflies, which demonstrated that barriers to introgression are concentrated in regions of low recombination [67,76], and supports a polygenic barrier architecture. This architecture was originally proposed as a null model of reproductive isolation [114] and evidence for polygenic barriers to gene flow has accumulated across a range of taxa [98,115–117].

We investigated whether barrier regions are associated with particular genomic features and failed to find enrichment of repetitive elements and more surprisingly, coding sequence, as one might expect if reproductive isolation is driven by selection on genes. Our circular resampling procedure controls for differences in gene and repeat density between chromosomes, so it is clear that the reduced gene density for barrier regions (circular bootstrap, $p$<0.001, S3 Fig) is not simply a consequence of the negative correlation between gene density and chromosome length and the fact that barrier density is higher for long chromosomes. However, it may well be that the effect of intra-chromosomal variation in recombination rate vastly outweighs the impact of coding sequence density.

### Limitations

To infer barriers to gene flow conservatively, we use the strictest possible threshold ($m_{e,i} = 0$) and quantified the false positive rate in a parametric bootstrap (see methods). Whilst this potentially excludes actual barriers with $m_{e,i} > 0$ at putatively neutral flanking regions — which are the basis/input of our inference — this minimises the false positive rate.

Numerous factors may contribute to biases in our results (see [35], and [67] for `gIMble` limitations). Given that direct estimates of recombination are not available for *Iphiclides*, we cannot directly quantify the degree to which recombination rate heterogeneity contributes to migration rate variation, nor account for the fact that uncertainty in estimates of both short and long-term barriers depends on the local recombination rate.

There may be scope to improve the power to detect weak gene flow through modelling more detailed demographic scenarios. Firstly, whilst it is very likely that the *Iphiclides* pair has undergone repeated rounds of separation and gene flow, we have fit a much simpler model that assumes a single continuous rate of migration. Secondly, we have modelled gene flow as unidirectional and inferred the most likely direction by comparing models. In reality, gene flow between these species is likely bidirectional but asymmetric, as estimates in other systems suggest [118–120]. Expanding the `gIMble` framework to include isolation with initial migration (IIM) and bi-directional gene flow may improve our power to model the build-up of genomic barriers [67,121].

The `diem` genome polarisation algorithm is designed to work with even small amounts of low quality data. There should therefore be few power limitations working with high quality genome scale data, and indeed estimates of hybrid indices, interspecific heterozygosity, and visualisation of admixture all have high precision [68]. Estimating admixture tract size distributions requires a further level of precision however: a single ancestry state error in a long block will on average halve its estimated length, error due to a miscalled variant would affect one tract, but error due to a mispolarised site could affect many. To avoid such error-driven tract fragmentation, we filter polarised sites for high diagnostic index — strongly correlated with the support for correct polarity — and introduce an arbitrarily chosen scale of kernel smoothing of state along chromosomes. This has an advantage over HMM approaches in that it has no starting point chirality, but the disadvantage that it censors true small tract signal with some distribution of false negatives. This, if anything makes, the excess of small tract observations over simple model expectations more surprising (Fig 6).

Finally, one of the most striking results of our analysis of HZ is the scarcity of introgressed ancestry tracts on the Z chromosome (Fig 3). While we have not quantified the density of `gIMble` barriers on the Z (because such an analysis would be limited to male samples), it will be fascinating to investigate the long-term evolution of Z linked barriers in *Iphiclides* in the future.

## Conclusion

We have demonstrated an association between long-term and short-term reproductive isolation in a species pair of swallowtail butterflies. Despite being several million generations old, gene flow persists between these species and is currently concentrated at a HZ. The considerable number, varying size, and location of barrier regions suggest that the genomic architecture of speciation is polygenic, or multilocus *sensu* Barton [88]. We find that variation in $m_e$ correlates with proxies of recombination rate variation both between and within chromosomes and so is likely shaped by recombination rate heterogeneity, a pattern previously observed in *Heliconius* butterflies [67,76]. However, in the absence of a directly estimated recombination map, it is impossible to know to what extent the overlap between long and short-term barriers reflects a shared/stable recombination landscape rather than shared selective targets. Unlike genomic cline methods [122], our approach maximises information about barriers to gene flow contained in small samples over different time scales. This paves the way for future quantitative analyses of the temporal evolution of the genomic landscape of species barriers. While most speciation processes are far too slow to observe directly, genomic variation clearly contains information about the interplay of forces acting at different stages of the speciation continuum.

## Methods

### Ethics statement

Field sampling of butterflies was conducted in compliance with the School of Biological Sciences Ethics Committee at the University of Edinburgh and the European Research Council ethics review procedure. Permissions for field sampling were obtained from the Generalitat de Catalunya (SF/639), the Gobierno de Aragon (INAGA/500201/24/2018/0614 to Karl Wotton) and the Gobierno del Principado de Asturias (014252).

## Sampling and sequencing

In total, we generated WGS data (150 base paired-end reads) for 20 individuals (S1 Table). Field sampling was conducted in 2017 and 2018 at several locations across Southern and Central Europe (Spain, France, Romania, and Hungary). Samples were hand-netted in the field, flash-frozen from live in a liquid nitrogen dry shipper (Voyageur 12) and stored at –70 °C. Wings were retained for identification. DNA extractions for all individuals were performed using a Qiagen DNeasy Blood & Tissue kit. Extractions were used to prepare TruSeq Nano gel-free libraries by Edinburgh Genomics which were sequenced on a NovaSeq 6000 or HiSeq X (S1 Table). Raw reads are deposited at the ENA (accession number PRJEB76171).

## QC, read mapping, variant calling, and summaries

Reads were trimmed and checked for quality using `FastQC v0.11.8` [123] both before and after trimming with `FastP v0.20.0` [124], using `MultiQC v1.7` [125] to visualise the results. Trimmed reads were aligned to the *I. podalirius* reference assembly [126] using `bwa-mem v0.7.17` [127]. We marked duplicates using `sambamba v0.6.6` [128]. Variants were called using `freebayes v1.3` [129] (`-k -w -j -T 0.02 -E -1 --report-genotype-likelihood-max`). A principle components analysis (PCA) was performed on intergenic autosomal variants using `plink v1.9` [69]. Genetic diversity ($\pi$), mean individual heterozygosity ($H$) and divergence ($d_{XY}$) were estimated at intergenic sites using the `gIMble` 'gimbleprep' module [67].

## Fitting models of speciation across the genome using the bSFS

To infer the likely speciation history of this species pair and to estimate migration rates along the genome we used the software package `gIMble` [67]. This analysis was restricted to the non-HZ set of individuals. As the inference assumes that each sample contributes diploid genotypes, analysis was restricted to the autosomes to allow the inclusion of both male and female data. Demographic models fit by `gIMble` assume a neutral model of evolution, and so we focused analysis on intergenic sequence (39% of the assembly), i.e. genic and repeat-rich regions and contigs that were not scaffolded into chromosomes were excluded. We filtered genotype (GT) calls to a minimum depth of eight reads per sample and a maximum depth of three times the mean coverage per sample. Additionally, GT calls were required to have a minimum PHRED quality of one and SNPs within 2 bases of indels were removed. We used the `gIMble` 'parse' module to quantify genetic diversity and divergence for the filtered subset of the data. We summarised variation in pair-blocks of 64 bases and tallied all blockwise site frequency configurations of mutations across heterospecific sample pairs (see [67] for details). We then fitted 'strict divergence' (*DIV*) and 'isolation with migration' (*IM*) demographic models to the genome-wide blockwise site frequency spectrum using the `gIMble` 'optimize' module.

To quantify heterogeneity in $m_e$ and $N_e$, we fit a grid of parameters to sliding windows of a fixed length of 28,125 (pair-blocks) with a 20% overlap along the genome. This results in a total of 14,173 windows with a minimum span of 50 kb. The grid discretised $N_e$ from 100,000 to 1,500,000 (in increments of 100,000) for *I. feisthamelii* from 100,000 to 900,000 (in increments of 50,000) for *I. podalirius* and from 100,000 to 2,400,000 (in increments of 100,000) for the ancestral population; $m_e$ (forwards in time from *I. feisthamelii* into *I. podalirius*) was discretised from 0 to 4e-7, in increments of 5e-9. The split time $T$ was fixed to the global result of 2,181,320 generations (Table 2) and the mutation rate was set to $\mu = 2.9 \times 10^{-9}$ per

base and generation [130] throughout. We label windows as barriers to gene flow where the marginal support for a *DIV* model (i.e. $m_e = 0$) is greater than the support for an *IM* model parameterised with the most likely inferred effective migration rate under a null model of $m_e$ variation (i.e. $\Delta_{B,0} > 0$).

## Simulations and bootstrapping

We used the following bootstrapping approach to test the global support for an *IM* model over a *DIV* model: we performed 100 simulations using `msprime` [131] (via the `gIMble` 'simulate' module) parameterised by the *DIV* history which best fit the empirical data. We simulated 14,173 windows of the same size as the real data, i.e. 28,125 (64 base) pair-blocks, for six diploid individuals per population to match the empirical sample set. We simulated a discrete genome and an infinite allele mutation model. We assumed a constant per base crossover rate based of one crossover per male meiosis per chromosome (i.e. each chromosome has a sex-averaged map length of 25 cM) and a mutation rate of $2.9 \times 10^{-9}$ per base and generation per base and generation, as estimated for *Heliconius* butterflies [130]. We fit a *DIV* model and an *IM* model to each simulation and compared the null distribution of relative fit (ln *CL*) to the empirical ln *CL*.

To adjust our barrier definition for false positives (which are expected at an appreciable rate when the background $m_e$ is low), we ran a parametric bootstrap on the local estimates $m_e$: for each window, we simulated 100 replicates under the best fitting local background history using `msprime` [131] (via the `gIMble` 'simulate' module) and obtained a null distribution of $\Delta_{B,0}$. These simulations were analogous to the bootstrap for the global model, except that we allowed $N_e$ parameters to vary between windows. We simulated under a null model with a globally fixed $m_e$, but accounted for variation in $N_e$ by assuming the best composite log-likelihood $N_e$ parameter inferred for each window. Only windows for which $\Delta_{B,0}$ in the real data was greater than the largest value contained in the $\Delta_{B,0}$ distribution from simulated data were labelled as barriers. This ensures a false positive rate <0.01.

To test whether various metrics are associated with barrier windows we used a simple resampling approach. In all instances, random empirical estimates of each per-window metric were sampled without replacement to generate datasets of sizes relevant to the scope of the resampling scheme. We compared the empirical estimate to the distribution of means of 1000 resampled data sets. Unless otherwise indicated, we resampled datasets with respect to each chromosome in a circularised fashion (similar to the shift-permutation approach outlined by Yassin et al. [71] and Nouhaud et al. [72]), by shifting the assignments of barrier windows within a chromosome irrespective of chromosome ends by a random integer. This approach accounts for between-chromosome variation in recombination rates and generates null distributions that have the same clustering as barriers inferred in the data. Finally, we repeated each circular bootstrap after excluding each of the HZ samples in turn to check for robustness to sub-sampling. For each of these tests we consider an adjusted significance threshold of $p = 0.05/6 \approx 0.008$. The bootstrapping was implemented using a custom script available at https://github.com/LohseLab/circular_bootstrap.

To test whether various measures of barrier strength are robust to heterogeneity in highly diagnostic marker density we employed a downsampling approach. SNPs were downsampled with over dispersion on the physical metric by choosing those with reference positions closest to multiples of 'spacing' 1100nt. Different spacing options were first trialed; 1100nt was the smallest spacing at which all chromosomes showed no significant deviation from uniformity after downsampling (Cramér-von Mises test, p=0.05). As downsampling cannot remove gaps,

and gIMble windows do not occur in SNV gaps, for each chromosome the 5 largest gaps in SNV positions were excised for the purposes of this Uniformity testing.

### Estimating hybrid indices and the ancestry of hybrid zone individuals

We polarized all variable sites (across all individuals) that were not singletons (which are uninformative for genome polarisation by association in state) and estimated genome-wide hybrid indices (HIs) using the `diem` framework implemented in both R and Mathematica [68]. Variants were subject to the same filtering criteria as the `gIMble` analysis. We calculated HIs per individual using sets of polarised sites with high diagnostic index (DI > -20). We used chromosome subsets of these sites to calculate HIs per chromosome. At a further reduction in scale we computed HI for each `gIMble` window and compared $D$ across all individuals between barrier and non-barrier `gIMble` windows using the bootstrap scheme described above.

### Deconfounding signal from multiple barriers

`diem` is designed to polarise genomes with respect to one barrier at a time, however trial `diem` analysis of the full genomes dataset suggested two barriers were present in the data: the focal barrier to gene flow between 'French' *I. podalirius* and 'Iberian' *I. feisthamelii*, and a 'nuisance' barrier to geneflow between 'Iberian' and 'African' *I. feisthamelii*, presumably due to the Strait of Gibraltar. For the focal analysis, signal of the nuisance barrier was removed by censoring all sites with minority homozygous state only present in (the two) samples from north Africa.

### Kernel smoothing of genotypes along chromosomes

`diem` estimates the polarity of each site independently. Where a barrier exists we may expect polarisation to reveal tracts of different inheritance along chromosomes. Downstream analyses of tract size distributions requires estimates of the boundaries of these tracts. We estimate tract boundaries using kernel smoothing of the diploid state along chromosomes. Working on the physical (Mb) metric, the Laplace distribution truncated at its 95% probability boundaries provides the kernel. This is centered on each site to provide a weighted average for each state 'smoothed' over the chosen physical scale. The centre-site is then estimated to have the state with greatest smoothed weight. Where variable sites are dense on the physical metric, smoothed estimates will be influenced by many flanking sites, where sparse, few. Imposing a given smoothing kernel scale assumes we are uninterested in state variation at higher frequency (involving shorter tracts). The kernel smoothing is therefore being used as a low-pass signal filter to leave larger tracts with clearer boundaries.

### Untangling blocks from genotypes

When diploid genomes are polarised with respect to a single barrier, an interval of heterozygous sites within otherwise homozygous chromosome genotype suggests a single tract of minority inheritance on one of the chromosome copies. While the same pattern would result from two minority tracts, one on each chromosome copy, that 'perfectly' abut (without overlap), simple genotypes can be parsimoniously (and automatically) parsed into haplotype tracts by assuming such perfect abutment is so rare as to be negligible. This accounts for the vast majority of the data: for each such heterozygous a block on one strand is counted. Occasional homozygous introgressed intervals are treated similarly: a

block on one strand is counted. Finally (and rarest of all), intervals of mixed homozygous-introgressed and heterozygous state are treated similarly: a block on one strand, spanning the mixed interval, is counted" In this way small blocks which positionally overlap with larger introgressing blocks within the same diploid individual are censored from the block size distribution.

## Supporting information

**S1 Appendix.**
(PDF)

**S1 Table.**
(CSV)

**S1 Fig.**
(PDF)

**S2 Fig.**
(PDF)

**S3 Fig.**
(PDF)

**S3 Fig.**
(PDF)

**S4 Fig.**
(PDF)

**S5 Fig.**
(PDF)

**S6 Fig.**
(PDF)

**S7 Fig.**
(PDF)

**S8 Fig.**
(PNG)

**S9 Fig.**
(PNG)

## Acknowledgments

We would like to thank Carla and Oskar Lohse for catching the two most informative hybrid samples: individuals 1322 and 1325. We thank Katy McDonald for help in the molecular lab, Alex Hayward for help with field collections and Edinburgh Genomics for generating libraries and sequence data. We are indebted to Vlad Dincă, Raluca Vodă and Leonardo Dapporto for contributing samples and to Nick Barton for helpful suggestions on the analyses of admixture tracts and Alex Mackintosh for insightful comments on an earlier version of this manuscript. We thank Richard Lewington for permission to reproduce his butterfly illustrations.

## Author contributions

**Conceptualization:** Sam Ebdon, Roger Vila, Stuart J. E. Baird, Konrad Lohse.

**Data curation:** Sam Ebdon, Dominik R. Laetsch, Konrad Lohse.

**Formal analysis:** Sam Ebdon, Dominik R. Laetsch, Stuart J. E. Baird, Konrad Lohse.

**Funding acquisition:** Sam Ebdon, Roger Vila, Konrad Lohse.

**Investigation:** Roger Vila.

**Methodology:** Sam Ebdon, Dominik R. Laetsch, Stuart J. E. Baird, Konrad Lohse.

**Project administration:** Sam Ebdon, Konrad Lohse.

**Resources:** Roger Vila.

**Software:** Dominik R. Laetsch, Stuart J. E. Baird.

**Supervision:** Dominik R. Laetsch, Stuart J. E. Baird, Konrad Lohse.

**Visualization:** Sam Ebdon, Stuart J. E. Baird, Konrad Lohse.

**Writing – original draft:** Sam Ebdon, Stuart J. E. Baird, Konrad Lohse.

**Writing – review & editing:** Sam Ebdon, Dominik R. Laetsch, Roger Vila, Konrad Lohse.

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
