## [Decision Letter · Decision Letter 0]

9 Aug 2024

Dear Dr Ebdon,

Thank you very much for submitting your Research Article entitled 'Genomic regions of current low hybridisation mark long-term barriers to gene flow in scarce swallowtail butterflies' to PLOS Genetics.

The manuscript was fully evaluated at the editorial level and by independent peer reviewers. The reviewers appreciated the attention to an important problem, but raised some substantial concerns about the current manuscript. Based on the reviews, we will not be able to accept this version of the manuscript, but we would be willing to review a much-revised version. We cannot, of course, promise publication at that time.

If you decide to revise the manuscript for further consideration at PLOS Genetics, please aim to resubmit within the next 60 days, unless it will take extra time to address the concerns of the reviewers, in which case we would appreciate an expected resubmission date by email to plosgenetics@plos.org.

To resubmit, log into your Editorial Manager account and select the option 'Revise Submission' in the 'Submissions Needing Revision' folder.

We are sorry that we cannot be more positive about your manuscript at this stage. Please do not hesitate to contact us if you have any concerns or questions.

Yours sincerely,

Nicolas Bierne

Academic Editor

PLOS Genetics

Kelly Dyer

Section Editor

PLOS Genetics

Dear Dr. Ebdon,

Thank you for submitting your manuscript to PLoS Genetics. It has been carefully reviewed by three referees. Overall, we appreciated your use of two innovative methods, gIMble and diem - one for demographic reconstruction and barrier locus mapping, and the other for chromosome painting in hybrids - within the same study. We also found your approach of comparing the results of these methods by analyzing parental populations far from the contact zone and hybrids within the contact zone interesting. In addition, we were intrigued by the result highlighted in your title, provided it was well supported. However, the reviewers have identified several concerns that need to be addressed before your manuscript can be considered for publication. Chief among these is the issue of small sample size, which was discussed by all of them. While the reviewers are open to the possibility that relevant conclusions can be drawn from a small sample, they remain unconvinced. About gIMble, while small sample size may not be a significant issue for inferring demographic parameters, it poses a challenge for barrier mapping. This aspect needs to be further explored and explicitly acknowledged in your manuscript. If your primary objective is to correlate the results with those obtained using diem, the effect of sample size may be less critical. However, it must be explicitly recognized that the positions of each individual barriers should not be considered strongly supported. Regarding the diem method, there is also a lack of clarity about the time scale to which your results apply. Given that your hybrid zone is likely partially congealed, and that only one individual could be considered an early generation hybrid, while the others are likely introgressed local parents, the congruence with the gIMble results may not be surprising. These local parents could simply be more introgressed, with bigger introgression tracts, compared to allopatric parents. The diem analysis again raises the question of whether your sample size is sufficient for conclusive results. In addition, you should acknowledge that your approach is conceptually similar, though not identical, to the genomic cline approach and discuss the relevant literature. It is important to explicitly address these issues while explaining that your primary goal is the correlation shown in Figure 6. You must also propose a solution to ensure that local errors do not unduly influence the overall correlation. Finally, as referee #2 pointed out, if the main objective of your study is the result shown in Figure 6, it needs to be discussed thoroughly. Was it expected or unexpected? How does it differ from the recent meta-analysis by Frayer and Payseur (https://doi.org/10.1093/evolut/qpae044) that found no correlation? It is important to discuss how your natural hybrids differ from lab-crossed hybrids in this context.

A small additional point. Figure 5 is not a De Finetti triangle and the concept of a genomic Wahlund effect is misleading. In an admixed population different loci will quickly drift or sweep to different allele frequencies and deviation from the line is more of an expectation than an exception. Please delete the dotted line. The difference between chromosomes is interesting, though, and underexplored. Some chromosomes support lower heterozygosity, what is their length? their density in barrier loci? what is their contribution to the result in figure 6?

I'll stop here; you have a lot of comments to address in the reviewers' reports. It is imperative that you address all concerns in a thoroughly revised version of your manuscript. While there is no formal "reject/encourage" decision at PLoS Genetics, I am assigning your manuscript a major revision. Please be aware that while your analyses are of interest, PLoS Genetics places a strong emphasis on the novelty of a study, and I am not yet sufficiently convinced in this regard after this round of reviewing. Therefore, it is crucial that your revised text meets the expectations of your title. Using gIMble and diem will not suffice. You will also need to convince the reviewers that your methods are unbiased and that your sample size is adequate - neither of which is currently guaranteed.

I look forward to reviewing your revised manuscript.

Best regards,

Nicolas

Reviewer's Responses to Questions

**Comments to the Authors:**

Reviewer #1: In this study, Ebdon et al. describe barriers to gene flow between two species of Iphiclides butterflies at two different timescales, including the first genetic description of a hybrid zone between these species. They find a substantial barrier to gene flow that appears similar at both timescales. It is rare and difficult to compare these two timescales, and it was not a foregone conclusion that these approaches would identify the same loci within the genome. This manuscript utilizes two exciting, recently-developed techniques to draw inferences from an impressively limited number of individuals. This study will be exciting to many within the speciation community who are limited by sample collection. Additionally, the comparison of the distribution of tract lengths to theoretical predictions is particularly exciting.

A limitation of this work is the lack of a detailed recombination map for these species. The authors have addressed this in several ways, but it would be helpful to have two further questions clarified within the text: 1) Are the predicted relationships between chromosome structure and recombination rate on lines 202-205 supported by any literature within this genus?; 2) How do these results relate to the expectation that tracts should be broken down more slowly in regions of lower recombination, and thus we may see minor parent ancestry reduced at loci farther away from a barrier locus than in a region of high recombination? While the authors have already spent significant space on addressing the issue of recombination, I think it is important as it is key to some of the interpretations and would be relevant for other researchers who are unable to obtain a detailed recombination map for their species.

Furthermore, it would be useful for the authors to discuss how the amount of divergence between the species reflects on the power to detect barriers along the genome, and if that could contribute to the observed similarities between the methods.

While this manuscript is well written, there are a few terms that require clarification when first introduced. In most cases, these terms are defined in the Methods (presented here following the Results), but not defined when first introduced in the Results. Examples include: “delta B0” on line 191, “focal barrier” on line 215, and the coupling coefficient terms on line 243. I recommend that the authors revisit the Results section and bring all necessary information forward from the Methods section, if it is to remain at the end of the manuscript.

Minor comments:

The authors should clarify the proportion of the genome that is reproductively isolated. The abstract says 33% and at line 193 it says 35%. If these numbers are not the same for a reason, it would be helpful to clarify.

On lines 134 and 135, should these be references to Figure 1 rather than Figure 2?

Figure 5 is referenced in the text before Figure 4.

Appendix equation A-3 appears to be missing a negative sign in the exponent.

Reviewer #2: This paper compares to analyses of barriers to gene flow between two Iphiclides species. The first major analysis uses a nice recent IM approach to quantify effective rates of migration across the genome. The second method uses genomes from 6 (or sometimes 3) hybrids to investigate barriers that act currently. A central conclusion is the partial concordance between long- and short-term barriers.

Overall, I found this an interesting but frustrating paper to review. On the positive side, the new methods (Figs. 4-5) are exciting, and results from the IM methods are also interesting and well applied. On the negative side, the whole paper felt hastily written, with many methods inadequately reported (many Figures lack important information in the legends for example). I also thought that the paper lacked a clear message. As such, it was difficult to tell which results were meant as interesting descriptive asides, and which were meant to contribute to a broader conclusion.

1. Overall message

The final sentences of both the abstract and the conclusions seemed disappointing at the moment. The "aims and objectives" section currently names software packages and the four questions are just cryptic descriptions of the analyses that follow. The reader needs to understand exactly why it is important to compare long-term and short-term barriers. What exactly could we conclude if the two sets of barriers were congruent vs. incongruent? What would an intermediate degree of congruence tell us? None of this was made clear either from the introduction or the aims.

2. Assumptions of the methods

The paper assumes that the reader is quite familiar with several previous papers (especially Barton and Gale 1993; Baird 1995; Laetsch et al. 2023, including e.g., the methods of parametric bootstrap). Currently, key quantities (S, R, D etc.) are not properly defined. I think more details of the methods should be reported throughout.

3. Congruence between the two sets of barriers

An interesting claim of the paper is the congruence between the long- and short- barriers. This is demonstrated by a weakish negative correlation between m_e and D (reported just as a summary statistic), and by an ANOVA on D between barrier and non-barrier windows, (Figure 6).

If I am correct that this is the central result of the paper, the reader needs to be better convinced of its importance and robustness. The p value of the ANOVA is reported, but without summary stats, sample size, or tests of its parametric assumptions. We also need to understand exactly how independent the short- and long-term analyses are. Naively, we might expect analysis of recent hybrids to be confounded if the taxa had already undergone extensive gene flow in the recent past. Is this a problem or not?

The conclusion also mentions possible confounding with recombination rate. I understand that a proper test might be impossible, but couldn’t you, for example, compare results for short- vs long chromosomes?

Small points:

1. “we label windows as barriers to gene flow if a DIV history (me = 0) has greater marginal support than an IM history assuming the best fitting genome-wide value of me “ Is this correctly described as “the strictest possible threshold” (p. 18), and might results be affected by adaptive introgression?

2.

“Genomic Wahlund effect” Are you suggesting an explanation here (e.g. that the reduced heterozygosity follows from spatial structure)? If so I would explain, if not, I would avoid the label.

3.

“The deficit of coding sequence is likely a consequence of the strong negative correlation between gene density and chromosome length (Pearson’s ρ = -0.421) on one hand and the strong positive correlation between barrier density and chromosome length on the other. These together may explain the strong negative correlation between gene and barrier density (Pearson’s ρ = -0.652), which is partially driven by the numerous small chromosomes with very high numbers of genes and small numbers of barriers.”

Couldn’t this be tested?

Reviewer #3: Review of "Genomic regions of current low hybridisation mark long-term barriers to gene flow in scarce swallowtail butterflies"

This manuscript takes an interesting approach to make a lot out of a relatively modest data set to ask an interesting question in speciation genomics - the relationship between short term and longer term permeability of the genome to introgression. In contrast to the more traditional approach of comparing divergent population pairs at different points on "the speciation continuum" this study compares populations of the focal species pairs at different portions of the range - such that the allopathic populations reveal long-term historical introgression, while sympatric populations reveal the short-term recent process of hybridization and introgression. This is an interesting approach, which will undoubtably inspire others to perform similar analyses in their systems. That said, I have numerous concerns about this study.

First - before asking their major motivating question, the authors make a solid approach to asking basic question about the system and inferring the history of speciation and gene flow. Notably they fit an IM model model to their allopatric samples and infer a history of ongoing gene flow. They also show that this history fits better than a simple model of pure divergence and that simple simulations under the best fitting demographic model of pure diverge do not confuse the IM inference. While this is reasonable, there are a few weaknesses here that must be noted, if not fully fixed. First the details of simulation are quite vague - e.g. the authors say the used gable simulate to run these simulations. However this is not particularly informative as looking back at the gIMble paper, it seems that gIMble simulate is simply wrapper for msprime. As such the authors should a. Reference msprime as this seems to rob msprime of a citation and b. Provide more details about genome architecture - did they model the genome structure of their focal species? Or was this a bunch of unlinked sites? Or was this a few chromosomes all 1 Morgan long? Was the mutation rate fixed or variable across chromosomes etc etc.Similarly, it appears that the DIV model assumes a clean and instantaneous spell rather than split from a structured ancestral population etc etc etc. All of these details have important influence on the interpretation of the fact that neutral demographic simulations with a pure split of divergence are not confused for IM model as a misspecified demographic model can lead to poor inference. Similarly linked selection can also generate signals confused for introgression, especially in the IM framework [e.g. Smith and Hahn 2024 https://academic.oup.com/genetics/advance-article/doi/10.1093/genetics/iyae089/7683793]. I'm not sure what to do with these criticisms, as the authors are using currently established standard best practices of the field, but I am concerned that such issues could lead to incorrect inferences downstream.

I have similar concerns about the interpretation of the block length distribution. The authors note some (but not all) of the complicating assumptions involved in going from a block length distribution (which they note is further enhanced by the lack of a genetic map) to an inference of selection. I don't think this analysis provides much support or anything, and it is probably best to remove it.

The most exciting result is - of course - the inferred correlation between long term and short therm barriers to introgression. This is an interesting result, but it was not well explained or motivated. Here are a few issues which require some though / attention:

First off the justification for using D - an indirect measure of admixture proportion rather than some more radiation approach was not well explained. Trying to read the minds of the authors, my intuition is that this reflects "mixture LD" (sense Falush et al 2003), and that the authors were worried that some more straightforward local ancestry deconvolution approach would miss admixed blocks on this short time scale. But this was neither stated nor supported / justified so it would help if the authors elaborated some.

Second - the stats are not well explained - e.g. What is the ANOVA model for comparing barrier and non-barrier loci- is this simply a t-test or is there more to the model? What is done about the non-independence of windows ? My hunch is that both D and m_{e,i} are autocorrelated across the genome, so this should be addressed. Additionally, the stats reporting was strange - in one case a correlation coefficient was provided without a measure of significance, in another case a p value was provided without a summary of the effect size etc...

Third - the biological question was unclear. Is this a question about the repeatability of key barrier loci, or a question about if the landscape of recombination itself is sufficient to generate correlations in local ancestry across time? Either way, the authors should better articulate their biological question, and either develop a parametric model that include variables such as chromosomes length, and position - -or develop some form of nonparametric matched permutation) to see if the correlation exceeds predictions off these simple correlates (although it could be that this model being inadequate simply reflects the limited info about the genome etc). This is particularly important because the authors find that simple genome features well predict local introgression.

Finally, the sample size here is remarkably small - only five samples from sympatry - and two with relatively little admixture. While it could be argued that making much of little data is a strength of the question and approach, I would like to have some sense of power, sample size necessary etc. While one genome provide a nice collection of coalescent genealogies and sometimes can be quite revealing for pop gen, one recently admixed sample does not. A stronger paper would include both some form of jacknifing at the individual level and some complimentary simulation.

**Have all data underlying the figures and results presented in the manuscript been provided?**

Reviewer #1: Yes

Reviewer #2: **No: **

Reviewer #3: None

PLOS authors have the option to publish the peer review history of their article (what does this mean?). If published, this will include your full peer review and any attached files.

Reviewer #1: No

Reviewer #2: No

Reviewer #3: No

---

## [Decision Letter · Decision Letter 1]

5 Dec 2024

PGENETICS-D-24-00612R1

Genomic regions of current low hybridisation mark long-term barriers to gene flow in scarce swallowtail butterflies

PLOS Genetics

Dear Dr. Ebdon,

Thank you for submitting your manuscript to PLOS Genetics. After careful consideration, we feel that it has merit but does not fully meet PLOS Genetics's publication criteria as it currently stands. Therefore, we invite you to submit a revised version of the manuscript that addresses the points raised during the review process.

Please submit your revised manuscript within 30 days Jan 04 2025 11:59PM. If you will need more time than this to complete your revisions, please reply to this message or contact the journal office at plosgenetics@plos.org. Please include the following items when submitting your revised manuscript:

We look forward to receiving your revised manuscript.

Kind regards,

Nicolas Bierne

Academic Editor

PLOS Genetics

Kelly Dyer

Section Editor

PLOS Genetics

Aimée Dudley

Editor-in-Chief

PLOS Genetics

Anne Goriely

Editor-in-Chief

PLOS Genetics

**Additional Editor Comments:**

Dear Dr Ebdon,

Your revised manuscript was examined by two of the three referees of the first round. The third did not respond to my request and I did not wish to invite a new referee. Both referees gave a positive feedback about this revised version. I share their positive evaluation. However, referee 1's concern about divergence time is shared by me, and it remains an important caveat of your paper. We are not convinced that your deductions and interpretations allow you to be so categorical in your conclusions about the contrast between two very different evolutionary time scales. Your poor knowledge of the study system, the limited number of demographic histories considered (2 pop IM model, no space, no variation of connectivity through time etc...), and above all the small number of individuals in a hybrid zone whose spatial structure and level of local introgression (the tails of the clines) are little known, do not allow you to assess whether the coincidence of regions resistant to introgression in and far from the hybrid zone is a surprise or not. Your junctions approach is the best you can get from this very small sample, and I very much appreciate it, but that's not the issue, the issue is more about the generality of the interpretations. So yes, you find figures, 100-200 generations in the ZH and 2M generations between the parents, but what? Does it allow you to believe that you are doing better than previous studies that have attempted the same investigation? I receive, and share, and so do referee 1, the criticisms of lab crosses studies, but at least we know what type of hybrids we're studying. And I'm not very happy that you only make negative comments about the low number of recombination generations, as if all this literature should be thrown away, rather than trying to have a constructive discussion. In fact, I don't know whether I prefer 1,000 lab F2s or 6 hybrids from a little-known HZ, be they studied with an elegant junction method. I could also well prefer a genomic cline analysis with less markers but many individuals and a nice transect. In your case, you have your own vision of your complex HZ hybrids, but in reality we don't know very much (the fact you misunderstood my concern that you might have locally introgressed parents and an early generation hybrid between these introgressed parents, proves you have a preconceived view). I don't share your certainties. That said, contrary to my conclusion in the first round, I think your methodological approaches deserve to be published in PloS Genet, and the comparison of the two approaches remains super interesting. However, I must ask you to make an effort to correct your certainties throughout your manuscript. You should work on the abstract and completely rewrite the author's summary. Throughout the manuscript, you must try to mitigate this very strong idea that you are comparing things that are as different as you say they are. If you have a secondary contact, even though the HZ introgression queues hybrids have recently incorporated new neutral heterospecific tracts, they are expected to have the same barriers as the parents. That's what people have done by comparing divergence islands and genomic clines in other studies. Your methods may well be more elegant, and that's where the novelty lies, not so much in the conceptual findings. I understand that my comment is somewhat at odds with my recommendation of the first round, but your revision and the very nice new analyses you've done, as well as your inability to really convince me of the significance of finding coincident barriers with the two methods, prompt me to ask you to tone that result down a bit in your revised version. Sorry for the back and forth.

I look forward to reading your revised version soon.

Best regards,

Nicolas Bierne

Minor comments :

- "vastly different timescales", "very recent signatures" "long-term signatures over the history of divergence" etc. please mitigate.

- L46 "speciation in the face of gene flow" → episodes of gene flow during speciation

- L58 "contemporary contact zones are likely one of many instances of secondary contact generated by drastic environmental changes" : I don’t understand what you mean here, probably too much than what you could.

- L 173 "An IM model supports a history of speciation with gene flow". What can an IM model support other than an IM model? Given you do not test secondary contact or varying level of geneflow with time, do not conclude more than what you can. I realy don’t mind that you use an IM model to map the barriers, but do not conclude too much about the history of divergence and gene flow. That a hybrid zone in the southwest of France would not be a secondary contact after post-glacial recolonization would surprise me a lot.

- Figure 4 "The dashed line indicates the expectation under Hardy-Weinberg equilibrium" Again, please delete this dashed line. If I make an effort to go your way, this line could be useful if you plot the population average (but you don’t really have a population sample here), that this average is close to the line, and you find a way to show how far the individual scatterplot deviates from it as a measure of HW and linkage disequilibrium (but the variance in the ancetry does the job well). When you compare different parts of a genome, it's against the genome average, but given that an individual genomic average may well deviate from the line due to linkage disequilibrium in HZ or selection and drift in an admixed population, I don't see the point of showing that expectation. Finally, as I said it previously, in an isolated admixed population that converged toward HWLE, one might expect individuals to be close to this line under neutrality and slow drift in large populations, but this is never the case, you always deviate from it (notwithstanding that theory predicts selection for heterozygosity and ancestry of the major or fitter parent). When an expectation is not useful, when it is not well explained and when it can mislead the reader because it is unusual, I don't see the point of showing this expectation. To put it briefly, I don't agree with you at all that it's a useful expectation.

**Journal Requirements:**

1) Please amend your detailed Financial Disclosure statement. This is published with the article. It must therefore be completed in full sentences and contain the exact wording you wish to be published. Please ensure that the funders and grant numbers match between the Financial Disclosure field and the Funding Information tab in your submission form. Note that the funders must be provided in the same order in both places as well.

**Reviewers' comments:**

Reviewer's Responses to Questions

**Comments to the Authors:**

Reviewer #1: In this study, the authors investigate short- and long-term barriers to gene flow between two species of Iphiclides butterflies. They use two recently developed techniques (gIMble and diem) to draw inferences from small sample size. Interestingly, they find that the barriers identified through the two approaches overlap significantly. They find that the Z chromosome has very little introgression. This study is exciting because such comparisons of different times scales are rare.

This is the second time I have reviewed this manuscript. I appreciate the authors’ efforts to make improvements, and I believe the manuscript has been strengthened as a result. I appreciate the new emphasis on the overlap between the short- and long-term barriers, which is a novel contribution of this manuscript. I particularly appreciate the authors’ addition of junction number as a metric and their new permutation approach, as it is exciting to see this theory applied to real data. Previous attempts at applying junctions to real data have looked at very different types of hybrid populations that made this comparison difficult (e.g. Lavretsky et al 2019, https://doi.org/10.1002/ece3.4981).

That being said, I’m not sure that the authors have addressed my concern about divergence. I was less concerned with the time of divergence and more concerned with the uneven pattern of differentiated markers along the genome. At line 330, the authors state that “sites with high diagnostic indices are more densely populated within barrier windows relative to a null resampled distribution.” Is there an issue of relative power along the genome? It would improve the paper for the authors to address this.

Minor comments:

1) I agree with the authors that the Frayer and Payseur paper has several additional complications brought on by the lab vs wild cross comparison (as was the purpose of their discussion), but it may also be worth noting that the time scales are different (where late-stage hybrids in that comparison may be closer to the short-term barriers considered here).

2) Around line 205, it would improve clarity to explicitly say that the loci used for the sliding windows are not the same as those used for fitting the global IM model above.

3) At lines 332-334, you say that the number of junctions should be negatively correlated with barrier strength. While I agree a reduction is expected, I think there is evidence that the strength of the reduction is modulated by several factors and not just selection strength (e.g. Hvala et al 2018, https://doi.org/10.1111/evo.13509). If this is based on specific theory, it would be good to cite it. This does not impact your conclusion, as you only show a reduction and not a correlation.

4) At lines 516-518, the authors state that their approach may undercount small blocks, but they actually observe more small blocks than expected by their model. Please clarify this point.

5) On lines 583 and 596, the authors reference different Keightley papers for the same mutation rate.

6) In the Methods section titled “Untangling blocks from genotypes,” the authors state that no phasing choice is necessary to determine the lengths of minor tracks of introgression. While this is true, a choice would still be required in most cases where minor ancestry is homozygous. The legend of Figure 6 suggests that that does not happen in most cases, and this seems reasonable given the relative proportions of admixture. However, it would be useful to clarify in the Methods both that most blocks were heterozygous, and what was done with homozygous blocks.

7) At the end of the Figure 5 legend, there should be a reference to plot F rather than plot E.

Reviewer #2: This manuscript is greatly improved. I like the new stats and especially the new circular randomization. The intro and discussion also do a much better job of contextualizing the work, and explaining why its questions are interesting. I am grateful to the authors for their constructive response to the earlier review.

I have two very very small comments:

1. There are lots of acronyms and their component single letters in this ms (D, Z, I, H, M, HI, DI, HZ, IM, IF, HWE, LD, CL etc.). I know they are all standard, but I would double check if any of the rarer ones could be removed (e.g. mt, x, HWE, FPR), or different notation used when similar symbols appear in the same paragraph (e.g. p. 9 has H, I, Z, HZ, HI, and IF).

2. p. 6: "using a pre-computed grid" Not sure what this meant.

**Have all data underlying the figures and results presented in the manuscript been provided?**

Reviewer #1: Yes

Reviewer #2: Yes

PLOS authors have the option to publish the peer review history of their article (what does this mean?). If published, this will include your full peer review and any attached files.

Reviewer #1: No

Reviewer #2: No

**Figure resubmission:**
---

## [Editor Report · Decision Letter 2]

14 Mar 2025

Dear Dr Ebdon,

We are pleased to inform you that your manuscript entitled "Genomic regions of current low hybridisation mark long-term barriers to gene flow in scarce swallowtail butterflies" has been editorially accepted for publication in PLOS Genetics. Congratulations!

Yours sincerely,

Nicolas Bierne

Academic Editor

PLOS Genetics

Kelly Dyer

Section Editor

PLOS Genetics

Aimée Dudley

Editor-in-Chief

PLOS Genetics

Anne Goriely

Editor-in-Chief

PLOS Genetics

Comments from the reviewers (if applicable):

**Data Deposition**

http://datadryad.org/submit?journalID=pgenetics&manu=PGENETICS-D-24-00612R2

**Press Queries**

---

## [Editor Report · Acceptance letter]

PGENETICS-D-24-00612R2

Genomic regions of current low hybridisation mark long-term barriers to gene flow in scarce swallowtail butterflies

Dear Dr Ebdon,

We are pleased to inform you that your manuscript entitled "Genomic regions of current low hybridisation mark long-term barriers to gene flow in scarce swallowtail butterflies" has been formally accepted for publication in PLOS Genetics! Your manuscript is now with our production department and you will be notified of the publication date in due course.

With kind regards,

Anita Estes

PLOS Genetics

On behalf of:
